# CamEdit: Continuous Camera Parameter Control for Photorealistic Image Editing

**Xinran Qin**[1],∗  **Zhixin Wang**[2],∗  **Fan Li**[2],  **HaoYu Chen**[3],
**RenJing Pei**[2],  **WenBo Li**[4],  **XiaoChun Cao**[1]†
[1]Shenzhen Campus of Sun Yat-sen University  [2]Huawei Noah's Ark Lab
[3]HKUST (Guangzhou)  [4]CUHK

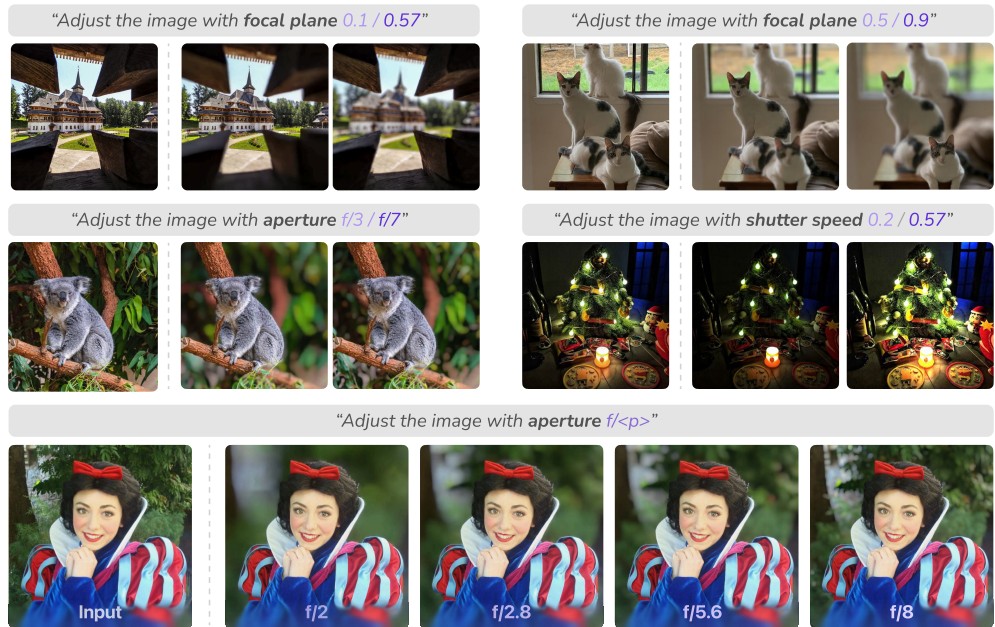

Figure 1: The proposed CamEdit enables photorealistic image editing through manual textual input of continuous camera parameters, including aperture, focal plane, and shutter speed, resulting in visually realistic outcomes.

## Abstract

Recent advances in diffusion models have substantially improved text-driven image editing. However, existing frameworks based on discrete textual tokens struggle to support continuous control over camera parameters and smooth transitions in visual effects. These limitations hinder their applications to realistic, camera-aware, and fine-grained editing tasks. In this paper, we present CamEdit, a diffusion-based framework for photorealistic image editing that enables continuous and semantically meaningful manipulation of common camera parameters such as aperture and shutter speed. CamEdit incorporates a continuous parameter prompting mechanism and a parameter-aware modulation module that guides the model in smoothly adjusting focal plane, aperture, and shutter speed, reflecting the effects of varying camera settings within the diffusion process. To support supervised learning in this setting, we introduce CamEdit50K, a dataset specifically designed for photorealistic image editing with continuous camera parameter settings. It contains over 50k image pairs combining real and synthetic data with dense camera

---

∗ Equal Contribution  † Corresponding Author

parameter variations across diverse scenes. Extensive experiments demonstrate that CamEdit enables flexible, consistent, and high-fidelity image editing, achieving state-of-the-art performance in camera-aware visual manipulation and fine-grained photographic control.

# 1 Introduction

Recently, diffusion models [1, 2, 20, 49, 50, 52, 54, 51, 32] have become powerful tools for both image generation and editing. They usually apply a pre-trained text encoder such as CLIP [48] and T5 [66] to inject manual textual prompts information into the generation process, enabling better generation quality and more precise control. Meanwhile, as social media platforms grow and smartphone cameras continue to advance, editing images to reflect photorealistic optical effects has become practically valuable. This highlights the need for editing methods that can directly manipulate camera parameters. However, most existing image editing methods [4, 5, 19, 23, 24, 29, 35, 58, 62] focus mainly on three main tasks: semantic editing, stylistic editing and structural editing.

Few prior works target photorealistic image editing, which edits indistinguishable from real photographs through precise control of camera parameters. In this work, we focus on the precise adjustments of focal plane[1] , aperture and shutter speed in camera parameters, which play a fundamental role respectively in determining focal range, background defocus degree, and exposure time [46, 57] during the photo-taking process.

Diffusion models capture strong spatial priors and scene geometry [55], making them suited for photorealistic editing. Recent works encode camera settings as discrete tokens within text-to-image (T2I) or text-to-video (T2V) generation frameworks [11, 70]. However, such discrete textual token-based approaches are difficult to directly apply to editing tasks involving continuous camera parameter control through textual prompt input (e.g. "*Adjust the image with aperture $f/2.8$*", etc.). This mismatch hampers smooth parameter adjustment and limits applicability to photographic editing.

To overcome these challenges, we introduce **CamEdit**, a diffusion-based framework for photorealistic image editing that allows continuous control of camera settings using text prompts. Instead of turning parameter values into separate tokens, we propose a continuous parameter prompting method, which interpolates between predefined anchor embeddings in the text space. This preserves alignment with representation distribution of the pre-trained model while enabling fine-grained control over a wide range of settings (such as "aperture $f/[2, 10]$", "shutter speed $[0, 1]$"). As diffusion backbones lack explicit camera priors and fail to capture parameter-specific effects, we further propose a *parameter-aware modulation* module that conditions spatial and channel features throughout the diffusion transformer, making explicit both local and global effects that text embeddings alone miss.

Given the lack of high-quality datasets for photorealistic camera-aware editing, we construct a hybrid dataset named **CamEdit50K**, which includes real-world photographs with extracted or estimated EXIF metadata[2], along with synthetic image pairs rendered under controlled variations in focal plane, aperture, and shutter speed. This dataset provides a strong foundation for learning models that are physically consistent and aware of the effect of varying camera parameters.

In summary, our main contributions can be summarized as follows:

- We propose **CamEdit**, a diffusion-based framework for photorealistic image editing that enables continuous and fine-grained control over intrinsic camera parameters such as aperture, focal plane, and shutter speed, entirely through manual textual prompts.

- We design a *continuous parameter prompting* mechanism and a *parameter-aware modulation* module to enable smooth and physically consistent control across varying camera settings.

- We construct a dataset, **CamEdit50K**, which contains aligned image pairs and corresponding camera parameter instructions, addressing the lack of supervised data for photorealistic image editing with continuous camera parameter.

---

[1]Focal plane is corresponding to the focal point in camera settings.

[2]EXIF is metadata embedded in image files that records camera settings such as aperture and shutter speed.

## 2   Related Work

**Image Editing with Diffusion Models.** Recent diffusion-based generative methods such as Imagen [54] and DALLE [49] leverage diffusion models conditioned on manual textual prompts to control the generation process. Consequently, Textual Inversion [14] and DreamBooth [53] allow for personalized image generation base on diffusion models. ControlNet [71] adds more control by using conditions like depth, edges, or pose. LoRA-based techniques [22, 34, 35] update only a small part of the model for quick adaptation. Unlike image generation, image editing modifies the style, structure, or content of an existing image to achieve specific goals. Prompt-to-Prompt [19] manipulates cross-attention between source and target prompts to guide the editing process, while InstructPix2Pix (IP2P) [5] extends this paradigm by fine-tuning diffusion models on synthetic triplets of (image, instruction, target). More recent approaches, such as InstructDiffusion [17] and MGIE [13], unify instruction-driven editing across a broad range of tasks and datasets, advancing general-purpose visual manipulation. To improve spatial fidelity, follow-up works incorporate localization priors such as masks and bounding boxes [58, 24, 4] to better preserve background consistency. Other efforts explore multi-task instruction tuning on large-scale synthetic datasets [37, 69, 64], enabling finer-grained semantic control. Approaches with sliders [16, 15] enable continuous control over the attributes of image edits. Despite these advances, existing frameworks remain centered on semantic and stylistic edits, which are rarely considering the growing needs for photorealistic editing.

**Camera-Aware Models.** Traditional camera-aware editing methods have shown strong performance in tasks such as focal plane adjustment [57, 45, 46, 63] and aperture simulation [7, 56]. These methods are typically based on physically inspired image formation models and often require additional inputs such as depth maps or aperture geometry [57, 45, 46, 18, 61, 63]. However, their applicability is generally limited to single-purpose editing scenarios due to their dependence on auxiliary data and restrictive modeling assumptions.

Recent diffusion-based approaches introduce camera control into generative pipelines, mainly focusing on extrinsic parameters like pose, viewing angle [8, 21, 33], or motion trajectory in text-to-video generation [40, 65, 68]. Conditioning is commonly achieved via camera tokens or scene descriptions to enable view synthesis and motion control. Some methods embed camera parameters as discrete tokens into T2I [11, 12] and T2V [70] diffusion models to generate images with varying physical properties. However, these approaches face two main limitations: they do not support editing real images in a physically consistent manner, and they represent continuous camera parameters in a discretized form, which restricts control precision and limits generalization.

## 3   CamEdit50K Dataset

Existing camera-aware datasets [11, 56, 70, 7] mainly focus on generation tasks, often lacking aligned image pairs or sufficient variation in camera parameters and content diversity. To address these limitations, we introduce CamEdit50K, a dataset specifically designed for photorealistic image editing under continuous, physically grounded camera control.

As shown in Table 1 and Figure 2, CamEdit50K unifies paired real and synthetic imagery, multi-parameter coverage, and explicit camera settings to support camera-aware editing and evaluation. Real photos supply rich content but often lack complete metadata. We recover missing parameters through a real-data parameter estimation pipeline. Synthetic images are produced with a synth-data rendering pipeline and come with ground-truth camera values, which enables accurate and dense supervision. By integrating these complementary sources, CamEdit50K delivers diversity and controllability, enabling continuous camera-parameter editing.

**Real-Data Param Estimation.** For the majority without metadata, we estimate these parameters using physically grounded methods: (i) *Focal Plane:* To ensure consistency across images, we define the focal plane within a normalized depth range $[0, 1]$, representing focus from far to near. Depth maps are predicted by `Depth Anything V2` [67] and normalized accordingly. The in-focus region is identified by comparing the target image with an all-in-focus input [60], and its mean depth is used as the estimated focal plane. (ii) *Aperture:* Since aperture primarily governs background defocus, we measure blur using an edge-based estimation [27]. The estimated defocus level is then converted to an effective aperture diameter via a simplified thin-lens model [44]. (iii) *Shutter Speed:* We estimate

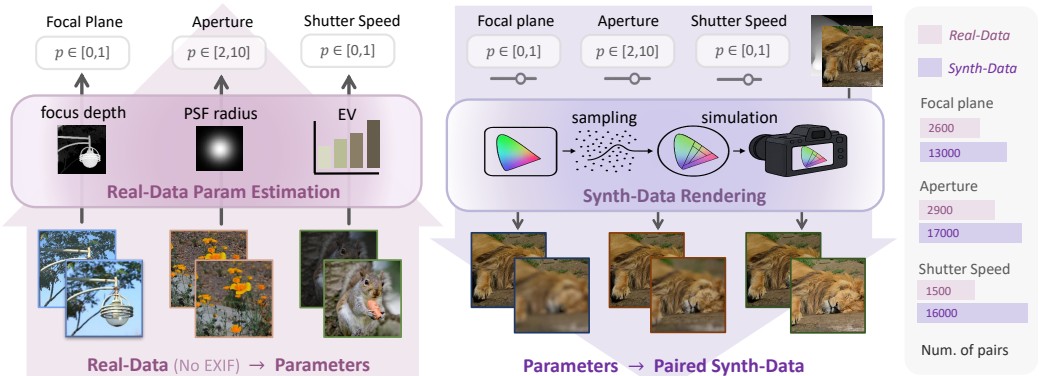

Figure 2: Overview of the CamEdit50K construction pipeline across focal plane, aperture, and shutter speed. Real image pairs without EXIF metadata are parameterized using physical cues such as focus depth, PSF radius, and exposure value. Synthetic pairs are generated by sampling camera settings and rendering photorealistic images under controlled conditions. The chart on the right shows the number of images for each parameter in CamEdit50K.

Table 1: Comparison with existing camera-aware datasets.

| Dataset | Venue | #Samples | Task | Real-Data | Synth-Data | Synth-Realism | Parameter-Dense | Scene Diversity |
|---------|-------|----------|------|-----------|------------|---------------|-----------------|-----------------|
| RealBokeh [56] | Arxiv 2025 | 23k | Render | ✓ | ✗ | - | ✗ | ✓ |
| Camera20k [11] | SA 2025 | 20k | Generate | ✓ | ✗ | - | ✗ | ✗ |
| PhotoGen [70] | CVPR 2025 | $3k^{n.3}$ | Generate | ✗ | ✓ | ✗ | ✓ | ✗ |
| **CamEdit50K** | – | 54k | **Edit** | ✓ | ✓ | ✓ | ✓ | ✓ |

exposure time from global brightness statistics and invert the camera response using a differentiable ISP pipeline [9, 31, 36], yielding a shutter speed consistent with the observed luminance.

**Synth-Data Rendering.** We generate synthetic image pairs by sampling camera parameters and rendering the corresponding effects. (i) *Focal Plane:* We sample the focal plane from the interval $[0, 1]$, selecting depths ranging from background to foreground. To simulate realistic depth-of-field effects, we employ a differentiable bokeh renderer [57]. (ii) *Aperture:* Using the depth map from [67], we fix the focal plane on the foreground and apply `BRIA.AI` [3] matting to preserve sharpness in the focused region. A thin-lens renderer [45] is then used to simulate varying aperture from $f/2$ to $f/10$, producing different degrees of defocus blur. (iii) *Shutter Speed:* We simulate exposure durations within the range $[0, 1]$ seconds by adjusting radiance in the HDR domain and converting it to RGB through a differentiable ISP pipeline [9, 31, 36].

## 4 Method

Our CamEdit framework adopts the instruction-driven editing paradigm of IP2P [5], while building upon the diffusion backbone as illustrated in Figure 3. Given a camera-parameter instruction, we first apply continuous parameter prompting as described in Section 4.1, which enables fine-grained prompt conditioning. The resulting embeddings, combined with the input image, are then fed into the transformer equipped with parameter-aware modulation modules described in Section 4.2, which inject parameter-specific feature modulation into the generation process.

### 4.1 Continuous Parameter Prompting

Directly learning parameter embeddings and appending them to other text features, while bypassing the text encoder, introduces a distributional mismatch with the frozen text embedding space. This misalignment degrades generation quality and hampers convergence, as shown in Section 5.3. To address this, our continuous parameter prompting synthesizes parameter representations by interpolating between anchor embeddings of adjacent discrete tokens within the text embedding space. The

---

[3]PhotoGen synthesizes samples by continuously sampling camera parameters over 3K fixed images, reducing scene diversity and affecting realism.

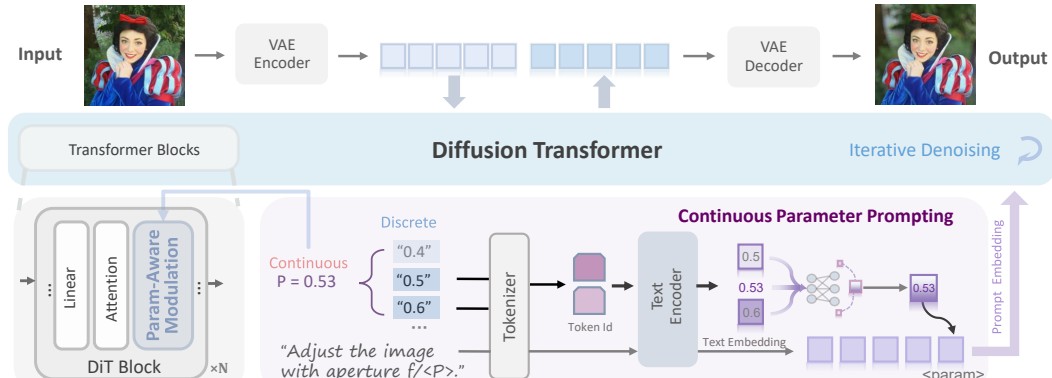

Figure 3: Framework Overview. The continuous parameter prompting module obtains a continuous parameter embedding via learnable interpolation over discrete camera token embeddings. This embedding replaces the placeholder token in the text embedding space, while preserving the original prompt structure. The parameter is also injected into the diffusion transformer via the parameter-aware modulation module, which adjusts features to reflect the corresponding visual effects.

resulting embedding replaces the parameter placeholder in the encoded prompt, ensuring perceptual continuity across parameter variations. This mechanism requires no modification to the text encoder and integrates seamlessly into diffusion-based frameworks.

Let $p \in \mathbb{R}$ denote a continuous camera parameter, and $\{p_1, \ldots, p_K\}$ be a set of predefined discrete anchor values with associated learnable embeddings $\{\mathbf{e}_1, \ldots, \mathbf{e}_K\} \subset \mathbb{R}^d$, where each $\mathbf{e}_k$ is obtained by encoding the anchor token via the frozen CLIP tokenizer and text encoder. For any $p \in [p_i, p_{i+1}]$, the parameter embedding is computed as:

$$\mathbf{e}_p = \text{Linear}([\mathbf{e}_i, \mathbf{e}_{i+1}]) + \text{MLP}(\phi(p)), \tag{1}$$

where $\phi(p) \in [0, 1]$ represents the normalized relative position of $p$ between anchors $p_i$ and $p_{i+1}$. The linear projection aggregates the semantic content of the two neighboring embeddings, while the MLP, implemented as a two-layer feed-forward network with ReLU activation, introduces a position-dependent residual to capture fine-grained variation. The final embedding $\mathbf{e}_p$ replaces the parameter placeholder in the encoded text prompt representation.

## 4.2 Parameter-Aware Modulation

Camera parameter variations primarily influence the spatial appearance of an image while preserving its underlying semantic content. Such changes include spatial transformations, for example, depth-dependent focus shifts across foreground and background regions [59]. Additionally, parameters such as shutter speed induce global exposure changes [30], motivating channel-wise feature modulation.

To effectively model visual effects, we modulate intermediate features conditioned on the parameter $p$ through two complementary operations as shown in Figure 4. To improve parameter sensitivity and information flow, the modulation is applied after the self-attention layers in each

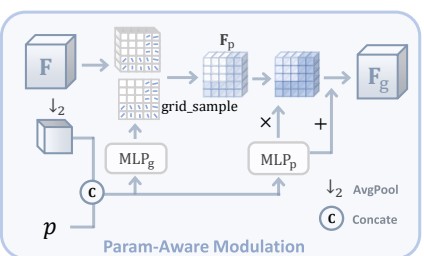

Figure 4: Illustration of parameter-aware modulation.

transformer block, where features contain rich contextual dependencies. The first component, geometry-aware spatial modulation, models lens-induced geometric distortion and depth-dependent focus transitions. It predicts a spatial displacement field that perturbs feature coordinates based on visual features and the input parameter. Specifically, we apply $2 \times 2$ average pooling to the input feature map $\mathbf{F}$ and feed the pooled representation, together with $p$, into a parameter-adaptive MLP:

$$\Delta \mathbf{G} = \text{MLP}_g(\text{AvgPool}(\mathbf{F}), p) \in \mathbb{R}^{2 \times H \times W}, \tag{2}$$

where `AvgPool`$(\cdot)$ denotes average pooling over non-overlapping $2 \times 2$ patches. The warped feature is then computed via:

$$\mathbf{F}_{\mathrm{g}} = \texttt{grid\_sample}(\mathbf{F}, \mathbf{G}_{\mathrm{base}} + \Delta\mathbf{G}), \tag{3}$$

where $\mathbf{G}_{\mathrm{base}}$ represents the canonical coordinate grid of $\mathbf{F}$ as defined in [25] and `grid_sample` performs differentiable sampling of $\mathbf{F}$ at continuous coordinates.

The second component, channel-wise modulation, adjusts feature amplitudes to capture global appearance variations. We compute channel-wise scaling and bias terms as follows:

$$\mathbf{F}_{\mathrm{p}} = \gamma(p) \cdot \mathbf{F}_{\mathrm{g}} + \beta(p), \quad \text{where } \gamma(p), \beta(p) = \mathrm{MLP}_{\mathrm{p}}(\texttt{AvgPool}(\mathbf{F}), p) \in \mathbb{R}^{C}. \tag{4}$$

Both $\mathrm{MLP}_{\mathrm{g}}$ and $\mathrm{MLP}_{\mathrm{p}}$ are implemented as single-layer feed-forward networks with intermediate ReLU activation. Then $\mathbf{F}_{\mathrm{p}}$ is forwarded to the subsequent transformer block.

## 5 Experiments

### 5.1 Implementation Details

**Training and Inference.** We adopt Stable Diffusion 3 (SD3) [10] as our backbone due to its strong generation quality and color fidelity. The majority of SD3 weights are kept frozen to preserve its pre-trained capacity. We update only the text embedding layer to learn anchor tokens and enable our learnable parameter interpolation. For each task, we predefine 10 anchor tokens via the tokenizer to guide training. The transformer is initialized from [72], and we fine-tune lightweight adapters using LoRA [22], combined with our physics-driven adaptation module. We train the model using AdamW [38], with learning rates of 1e-5. Training is conducted on $512 \times 512$ resolution images with a batch size of 32 for 50 epochs. During inference, the model requires only an input image and an instruction specifying any continuous camera parameter value within the valid range.

**Metrics.** We evaluate performance from three perspectives: perceptual quality, content preservation, and parameter control accuracy. Perceptual quality is measured using NIQE [42] and MUSIQ [28], which assess naturalness and visual fidelity without references. Content preservation is quantified via DINO similarity [6], capturing semantic alignment between the source and edited images. To assess parameter control, we compute the L1 error between the instruction-specified target and the estimated parameter extracted from the generated image. Estimation follows the physically grounded procedure described in Section 3. We evaluate 200 images across varying parameter settings.

### 5.2 Comparison to State-of-the-Art Methods

**Comparison Methods.** We firstly compare our method against state-of-the-art diffusion-based baselines, including editing models such as SuperEdit [41], UltraEdit [72], and In-Context Edit [47], as well as the camera-aware I2V model PhotoGen [70]. To evaluate PhotoGen [70], we generate images using prompts sampled from GPT-4o to simulate realistic text-based generation requests. We retrain UltraEdit on our CamEdit50K to enable camera-aware editing, denoted as UltraEdit*.

Beyond diffusion-based baselines, we further compare our method with other approaches across all editing tasks. For aperture editing, we compare with BokehMe [45], BRVIT [43], and DrBokeh [57]. For focal plane editing, we evaluate against BokehMe [45], MPIB [46], and DrBokeh [57]. For shutter speed editing, we include advanced low-light and exposure-aware methods such as SCI [39], CycleR2R [36], and CLODE [26]. All methods are tested under a consistent exposure configuration and evaluated for their ability to adapt image brightness while preserving both structural integrity and perceptual quality.

**Quantitative Comparison.** As shown in Table 2 (a), our method consistently outperforms other editing across all tasks in NIQE, DINO, and control error. Compared to the retrained UltraEdit* model, our method achieves an over 40% relative reduction in average control error, demonstrating substantially improved parameter alignment. Existing editing models do not explicitly incorporate camera parameters, limiting their ability to generate parameter-consistent results. Although UltraEdit* benefits from CamEdit50K supervision, it remains less precise than our approach.

We further evaluate performance via GPT-4o across photographic realism, content preservation, and parameter accuracy. We further assess performance using GPT-4o evaluation and a user study with 15 photography experts. Each participant evaluated 20 image sets per task across the three tasks, scoring

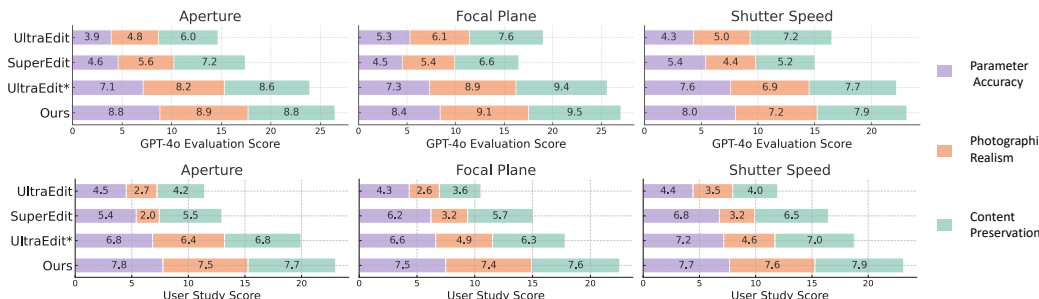

Figure 5: GPT-4o evaluation and user study across three dimensions: photographic realism, content preservation, and camera parameter accuracy, using a 0–10 scale (higher is better).

Table 2: Quantitative comparison across camera-aware editing methods under aperture, focal-plane, and shutter-speed control. Best results are in **bold**.

**(a) Diffusion-Based Methods**

| Method | Aperture | | | Focal Plane | | | Shutter Speed | | |
|---|---|---|---|---|---|---|---|---|---|
| | NIQE↓ | DINO↑ | Error↓ | NIQE↓ | DINO↑ | Error↓ | NIQE↓ | DINO↑ | Error↓ |
| SuperEdit [41] | 4.43 | 0.73 | ∼5 | 5.28 | 0.73 | ∼0.5 | 4.86 | 0.75 | ∼0.5 |
| UltraEdit [72] | 4.58 | 0.70 | ∼5 | 5.57 | 0.74 | ∼0.5 | 5.02 | 0.72 | ∼0.5 |
| In-Context Edit [47] | 4.30 | 0.78 | ∼5 | 4.91 | 0.80 | ∼0.5 | 4.29 | 0.81 | ∼0.5 |
| PhotoGen [70] | 6.65 | – | 1.31 | 6.21 | – | 0.28 | 6.11 | – | 0.19 |
| UltraEdit* [72] | 4.21 | 0.78 | 1.87 | 5.14 | 0.79 | 0.25 | 4.69 | 0.88 | 0.23 |
| **Ours** | **3.34** | **0.83** | **0.60** | **4.46** | **0.82** | **0.15** | **4.28** | **0.93** | **0.11** |

**(b) Other Methods**

| Method | Aperture | | | | Method | Focal Plane | | | |
|---|---|---|---|---|---|---|---|---|---|
| | NIQE↓ | MUSIQ↑ | DINO↑ | Error↓ | | NIQE↓ | MUSIQ↑ | DINO↑ | Error↓ |
| BokehMe [45] | 4.62 | 59.70 | 0.82 | 0.62 | BokehMe [45] | 5.19 | 48.97 | 0.79 | 0.22 |
| BRVIT [43] | 6.53 | 50.72 | 0.71 | – | MPIB [46] | 5.14 | 49.05 | 0.78 | 0.18 |
| DrBokeh [57] | 3.81 | 62.26 | 0.81 | **0.58** | DrBokeh [57] | 5.07 | 47.59 | 0.80 | 0.17 |
| **Ours** | **3.34** | **62.91** | **0.83** | 0.60 | **Ours** | **4.46** | **52.64** | **0.82** | **0.15** |

photographic realism, content preservation, and parameter accuracy. As shown in Figure 5, our method achieves the highest average scores on all dimensions. Relative to UltraEdit*, our CamEdit improves realism by 8% and parameter-control accuracy by 10%, demonstrating consistently higher realism, stronger content preservation, and more precise control.

As shown in Table 2 (b), our method consistently outperforms rendering-based baselines on most metrics, such as DrBokeh [57], BokehMe [45] on both aperture and focal plane editing tasks. For aperture editing, our approach yields 23% lower control error compared to baselines. Relative to the strongest competitor, DrBokeh, our method improves NIQE by 12% and reduces error by 24%. These results highlight the benefit of continuous parameter control in diffusion models, enabling physically grounded and perceptually faithful image editing.

**Qualitative Comparison.** As shown in Figure 6, our method achieves fine-grained and continuous control across all camera parameters, demonstrating clear parameter awareness. Existing diffusion-based editing models lack such a capability due to the absence of camera supervision during training. UltraEdit*, retrained on our dataset, shows improvement, but its discrete prompting leads to occasional mismatches when interpolating unseen values. In contrast, our method ensures smooth transitions and better fidelity, particularly around depth-sensitive regions such as foreground boundaries, owing to our parameter-aware modulation. We also compare with the generative model PhotoGen [70], where our results exhibit higher photographic realism and more coherent spatial structure. This improvement stems from our editing-based formulation and the use of real image pairs during training. Our method handles diverse scenarios with realistic parameter effects, such as light flares from aperture adjustment in nighttime scenes or exposure refinement that enhances visual aesthetics.

Figure 7 further supports our findings. Under aperture variation, our method preserves fine details like hair strands and object contours. For focal plane editing, we maintain sharpness in in-focus areas,

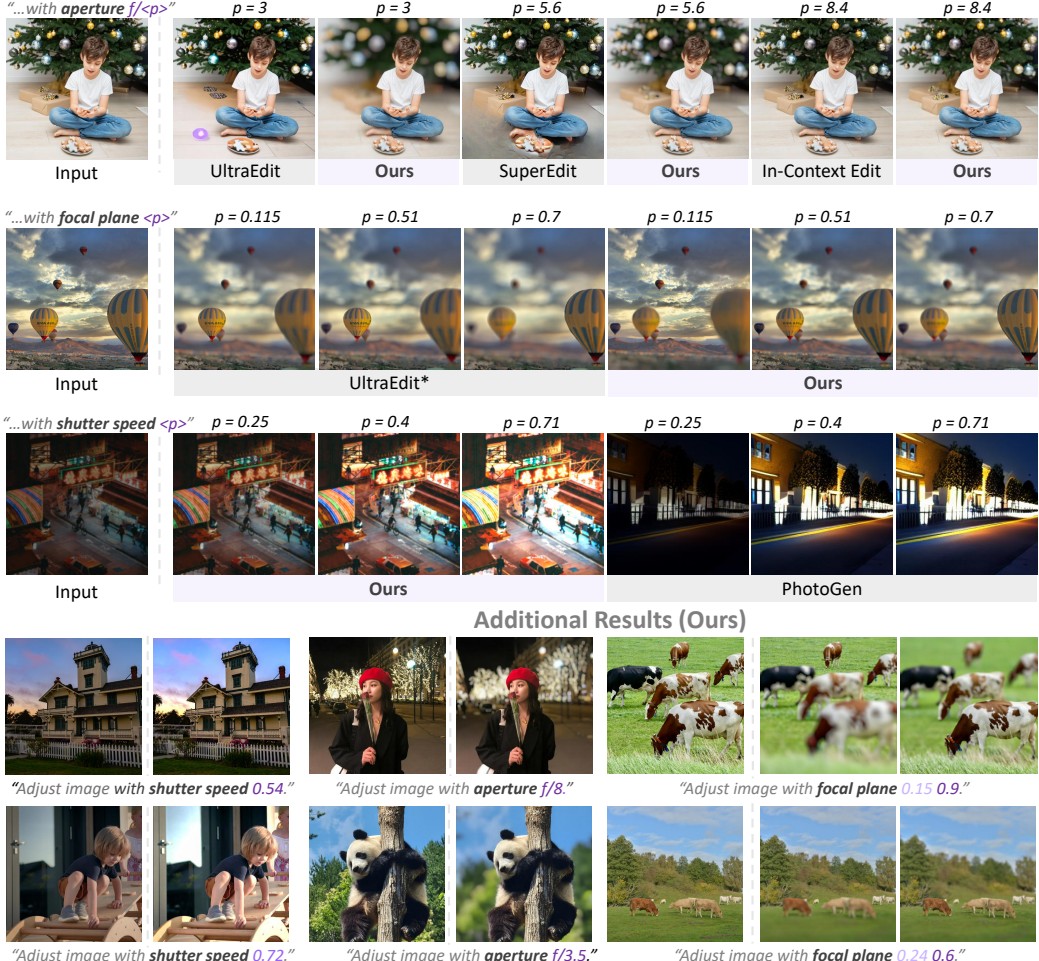

Figure 6: Visual comparison with diffusion-based methods. Our results illustrate smooth and perceptually consistent edits under various instructions.

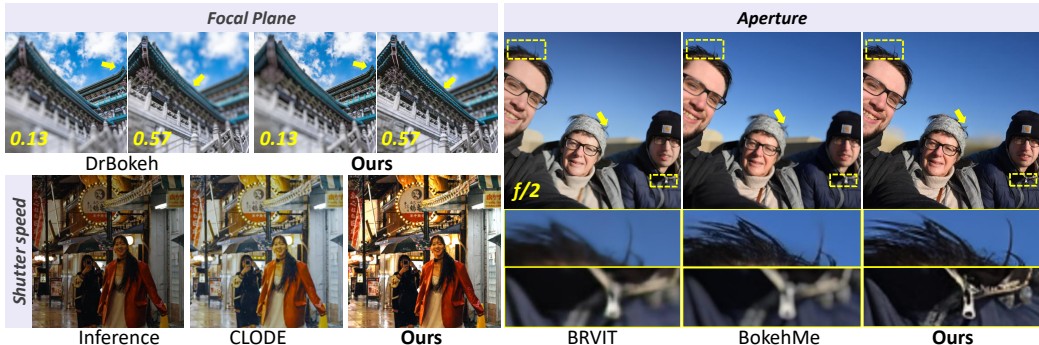

Figure 7: Visual comparison with other methods. Our method achieves finer detail and more realistic photographic effects compared to traditional rendering pipelines.

especially on architectural edges. In shutter speed control, our outputs adjust motion-related brightness while preserving photographic style. These results confirm that CamEdit delivers physically consistent edits with precise control and high visual fidelity.

## 5.3 Ablation Study

In this section, we analyze the components of CamEdit and the composition of CamEdit50K. All ablations are conducted on the focal-plane task.

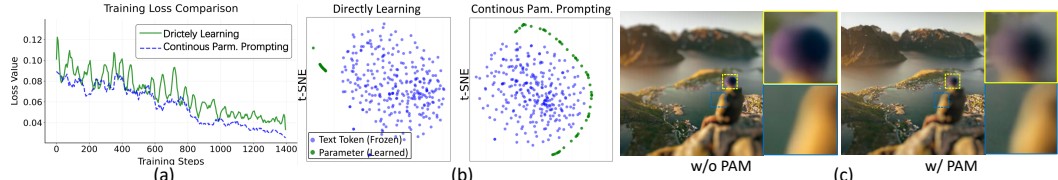

Figure 8: (a) Training loss comparison between direct parameter embedding and ConPrompt. (b) T-SNE visualization showing more semantically aligned embeddings with ConPrompt. (c) Visual comparison with and without PAM.

Table 3: Ablation study on key components of our CamEdit.

| Method | NIQE↓ | MUSIQ↑ | DINO↑ | Error↓ |
|---|---|---|---|---|
| w/o ConPrompt | 4.31 | 54.26 | 0.80 | 0.25 |
| w/ Direct Embed. | 5.25 | 52.41 | 0.81 | 0.27 |
| w/o PAM | 4.79 | 54.10 | 0.82 | 0.18 |
| **Ours (Full)** | **4.46** | **52.64** | **0.83** | **0.15** |

Table 4: Performance with different synthetic-to-real ratios of training dataset.

| Syn:Real | NIQE↓ | MUSIQ↑ | DINO↑ | Error↓ |
|---|---|---|---|---|
| 0 : 1 | 4.81 | 53.73 | 0.75 | 0.23 |
| 1 : 1 | 4.76 | 53.92 | 0.80 | 0.21 |
| 1 : 0 | 4.93 | 53.14 | 0.76 | **0.15** |
| **CamEdit50K** | **4.46** | **52.64** | **0.83** | **0.15** |

**Continuous Parameter Prompting.** As shown in Table 3, removing continuous parameter prompting (w/o ConPrompt), leads to degraded control accuracy and lower perceptual quality. Similar inconsistencies are observed in the retrained UltraEdit*, where editing results lack smooth transitions and do not align well with the target parameters, as illustrated in Figure 6.

We further evaluate a direct parameter embedding learning variant (w/ Direct Embed.), which bypasses the text encoder and learns parameter embeddings independently. It results in lower visual quality, and unstable training, as shown by higher loss and slower convergence in Figure 8 (a) and (b). In contrast, ConPrompt interpolates between pre-defined anchor tokens within the frozen text space, yielding smooth, semantically meaningful embeddings that improve control, fidelity, and stability.

**Parameter-Aware Modulation.** Parameter-aware modulation improves both parameter accuracy and image quality, as evidenced by the performance drop in the "w/o PAM" variant in Table 3. This is because different camera parameters induce global shifts in scene appearance, and PAM enables the model to adapt feature representations accordingly. As shown in Figure 8 (c), removing PAM results in unnatural transitions in defocus regions, particularly around human silhouettes, where the blur at object boundaries becomes abrupt. We also evaluate the injection of continuous camera-parameter features into the diffusion timestep embeddings in place of PAM. Compared with CamEdit, NIQE is higher by 0.30, DINO is lower by 0.03, and Error is higher by 0.09, indicating weaker parameter control due to the absence of localized spatial modeling.

**Composition of CamEdit50K.** We analyze CamEdit50K by varying the ratio of synthetic to real data, as shown in Table 4. Training on real data alone lacks scene diversity and parameter coverage, leading to weaker perceptual quality and control. Adding synthetic data at a 1:1 ratio improves over real-only training, though gains are limited by the smaller total size. Synthetic-only training scales well and enhances control, but visual fidelity lags without real-image guidance. Our CamEdit50K, combining available synthetic and real data, delivers the best parameter control and visual quality, driven by the scale of synthetic data and the fidelity of real data.

## 6 Conclusion

We present CamEdit, a framework for photorealistic image editing with continuous control over camera parameters such as aperture, focal plane, and shutter speed. It features a parameter-aware design and is supported by CamEdit50K, a hybrid dataset with paired images and varying camera settings. CamEdit enables visually consistent, photorealistic edits and lowers the barrier for users to manipulate camera parameters through images, with potential applications in education, simulation, and creative industries. While effective on key controls, it does not yet support all camera parameters and cannot recover focus from blurred regions, which remains fundamentally challenging and presents a valuable direction for future research.

### 6.1 Acknowledgments

This work was supported in part by the Shenzhen Science and Technology Program (No. KQTD20221101093559018), the National Natural Science Foundation of China (No. 62025604), and the CIE–Smartchip Research Fund (No. 2024-08). We gratefully acknowledge the creators and maintainers of the public datasets and open-source models used in this work.

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
