# OpenReview forum: "CamEdit: Continuous Camera Parameter Control for Photorealistic Image Editing"
_NeurIPS.cc/2025/Conference — NeurIPS 2025 poster_

### Official Review · Reviewer_U816 · 2025-06-24

**Clarity:** 3
**Significance:** 3
**Originality:** 3
**Rating:** 4
**Confidence:** 4

**Summary:**

This paper introduces a novel diffusion-based framework designed for photorealistic image editing through continuous control of camera parameters. Two core technical contributions are proposed: a continuous parameter prompting module which generates continuous parameter embeddings and a parameter-aware modulation module which modulates intermediate features conditioned on the camera parameter. Besides, a large dataset CamEdit50K combining real-world photos with estimated parameters and synthetic data rendered with precise ground-truth settings are introduced. Quantitative and qualitative results are provided to demonstrate the model's performance.

**Questions:**

- Does the three different camera parameters editing require training three different models?
- Could the author provide more quantitative results on a larger benchmark or testing set?
- More exploration and discussion about the current disability of inverse recovery are needed.

**Ethical Concerns:**

["NO or VERY MINOR ethics concerns only"]

**Final Justification:**

The authors have addressed most of my concerns about the Joint Parameter and the Benchmark in their rebuttal. Therefore, I maintain my score and recommendation for acceptance.

**Limitations:**

yes

**Paper Formatting Concerns:**

I did not notice any major formatting issues.

**Quality:**

3

**Strengths And Weaknesses:**

Strengths:
- The paper is well-motivated with a focus on a clear and important limitation in the current state of diffusion-based image editing.
- The attempt to construct parameter guidance on both parameter prompting and latent modulation is novel.
- A large dataset CamEdit50k is introduced in this paper, which is contributed to the community.
- The experimental results show that the proposed method surpasses previous methods.

Weaknesses:
- As mentioned in line 194, if I understand correctly, the three different camera parameters editing requires training three different models, which makes the method less practical and inefficient in the co-adjustment of different camera parameters.
- As mentioned in line 208, the evaluation is conducted on 20 images, which weakens the statistical reliability of the result.
- As mentioned in line 282, the proposed method "cannot recover focus from blurred regions". I’d recommend including more in-depth discussions on the current disability of inverse recovery (e.g., is the editing itself an irreversible process, or is it affected by the introduced modulation?).

---

> ### Author Rebuttal · Authors · 2025-07-31
>
> We sincerely appreciate your positive feedback and comments. Below are our responses to your concerns:
> > W1&Q1: Concern about whether editing three camera parameters requires training three separate models.
>
> We conduct an experiment training a unified model to adjust all three parameters simultaneously. In this setting, we doubled the number of training parameters and incorporated 5,000 synthetic images with multi‑parameter variations. As shown in Table R4‑1, the unified model achieves accuracy and output quality comparable to single‑parameter control, confirming its effectiveness in handling multi‑parameter adjustments. We will include these updated results and further clarify this capability in the revised manuscript.
>
> Table R4-1: Performance of CamEdit with single-parameter vs. joint multi-parameter control, evaluated across 200 images with various parameter settings.
>
> | Task             |           **Aperture** |          |          |           **Focal Plane** |          |          |           **Shutter Speed** |          |          |
> |--------------------|-----------------------------------:|---------:|---------:|-------------------------------------:|---------:|---------:|---------------------------------------:|---------:|---------:|
> |                    | NIQE ↓                            | DINO ↑   | Error ↓  | NIQE ↓                              | DINO ↑   | Error ↓  | NIQE ↓                                | DINO ↑   | Error ↓  |
> | **CamEdit (Single-Parameter)** | 3.34         | 0.83   | 0.60   | 4.46             | 0.84   | 0.15   | 4.28               | 0.93   | 0.11   |
> | **CamEdit (Joint-Parameter)** | 3.42         | 0.84   | 0.65   | 4.70             | 0.83   | 0.17   | 4.12               | 0.91   | 0.15   |
>
> > W2&Q2: Request for results on a larger benchmark.
>
> We expand the testing set to 200 images. As shown in Table R4‑2, CamEdit consistently outperforms all baselines across aperture, focal plane, and shutter speed, achieving superior perceptual quality, semantic consistency, and parameter accuracy on all tasks. These results are consistent with those reported in the main paper and will be incorporated into the revised version.
>
> Table R4‑2: Quantitative comparison of 200 images across aperture, focal plane, and shutter speed.
>
> | Method             |           Aperture            |          |          |           Focal Plane           |          |          |           Shutter Speed           |          |          |
> |--------------------|-----------------------------------:|---------:|---------:|-------------------------------------:|---------:|---------:|---------------------------------------:|---------:|---------:|
> |                    | NIQE ↓                            | DINO ↑   | Error ↓  | NIQE ↓                              | DINO ↑   | Error ↓  | NIQE ↓                                | DINO ↑   | Error ↓  |
> | UltraEdit          | 4.58                              | 0.70     | ~5.00    | 5.57                                 | 0.74     | ~0.50    | 5.02                                   | 0.72     | ~0.50    |
> | SuperEdit          | 4.43                              | 0.73    | ~5.00    | 5.28                                 | 0.73     | ~0.50    | 4.86                                   | 0.75     | ~0.50    |
> | UltraEdit*         | 4.21                              | 0.78     | 1.87     | 5.14                                 | 0.79     | 0.32     | 4.69                                   | 0.88     | 0.23     |
> | **CamEdit** | **3.34** | **0.83** | **0.60** | **4.46** | **0.84** | **0.15** | **4.28** | **0.93** | **0.11** |
>
> > W3&Q3: More exploration and discussion about the current disability of inverse recovery.
>
> We clarify that our method—similar to most camera parameter rendering approaches (e.g., DrBokeh [1], MPIB [2]) and image editing frameworks (e.g., SuperEdit [3])—is not designed to restore information that is already absent from the input. CamEdit with the proposed modulation simulates the effects of adjusting camera parameters but does not attempt to reconstruct content that was never captured. This limitation arises because our task focuses on photographic editing, where fidelity to the original content is prioritized over new content generation.
>
> However, we recognize the value of exploring inverse image restoration for image editing. Our CamEdit50K pipeline provides a foundation for such research, and we plan to investigate extending our framework for restoring degraded content in future iterations.
>
> [1] Yichen Sheng et al. Dr. Bokeh: Differentiable Occlusion-Aware Bokeh Rendering. CVPR, 2024.
>
> [2] Juewen Peng et al. MPIB: An MPI-based bokeh rendering framework for realistic partial occlusion effects. ECCV, 2022.
>
> [3] Fan Chen et al. SuperEdit: Rectifying and Facilitating Supervision for Instruction-Based Image Editing. ICCV, 2025.

---

> > ### Comment · Reviewer_U816 · 2025-08-04
> >
> > Most of my concerns have been addressed after reviewing the authors' rebuttal. I will maintain my score. I hope the discussion in the rebuttal concerning the Joint Parameter and the Benchmark will be incorporated into the revised manuscript to provide necessary clarification for readers.

---

> > > ### Author Response · Authors · 2025-08-04
> > > **To Reviewer U816**
> > >
> > > We sincerely thank you for carefully reviewing our response and greatly appreciate your recognition of our efforts to address the concerns raised. We will incorporate the joint parameter and the benchmark into the revised manuscript in response to your valuable suggestions.

---

> ### Author Response · Authors · 2025-08-04
> **To Reviewer U816**
>
> Dear Reviewer  U816,
>
> We sincerely appreciate your positive feedback, and have addressed your concerns accordingly. As the second review phase nears its end, please let us know if anything remains unclear or if further clarification would be helpful.

---

### Official Review · Reviewer_DyLt · 2025-06-29

**Clarity:** 3
**Significance:** 4
**Originality:** 4
**Rating:** 4
**Confidence:** 4

**Summary:**

The authors aim to address fine-grained photorealistic image editing problems. To this end, they propose a diffusion-based editing framework consists of continuous parameter prompting and parameter-aware modulation module to achieve this task. They also propose a  training dataset with 50k samples to facilitate model training. Extensive experiments shows the effectiveness of the method and each modules.

**Questions:**

I d like the authors to address the concerns about the dataset settings and evaluation methods part in the ''weakness'' part.

**Ethical Concerns:**

["NO or VERY MINOR ethics concerns only"]

**Final Justification:**

Most of my concerns have been addressed.

**Limitations:**

The authors should discuss limitations about the work.

**Quality:**

3

**Strengths And Weaknesses:**

strengths:
1. The target problem is interesting and significant to both image editing area and real applications. Existing manipulations cannot achieve fine-grained photorealistic image editing.
2. The paper is well written and organized. The main method is easy but effective, with the solid ablation studies. Also the proposed dataset will be a great contribution.

weakness:
1. some typos should be checked. (e.p line 184 G_base.)
2. The authors introduce the motivation of continuous parameter prompting and parameter-aware modulation in Sec 4.1 and Sec. 4.2. However, I d like the author to expose more ideas about how to link the proposed methods into the problem.
3. The proposed dataset is a contribution. Thus, the authors need to give more discussions on it. For example, the authors take the two ways to get the data. So, whats the pros and cons of each method? Will different data ratio leads to different results. Moreover, the scale of dataset will also be a factor to the results.
4. I also doubt the evaluation parts since the authors only use GPT-generated samples. so is it reliable? The better way is to make samples by human beings (like PS) to get real photo pairs.

---

> ### Author Rebuttal · Authors · 2025-07-31
>
> We sincerely appreciate your positive feedback and comments. Below are our responses to your concerns:
> > W1: About Typos.
>
> We will correct all typos in the revised manuscript.
>
> > W2: Clarification on how continuous parameter prompting and parameter-aware modulation connect to the problem.
>
> Existing diffusion-based editing frameworks face two key limitations when applied to photorealistic, parameter-controllable image editing:
>
> 1. Camera parameters are inherently continuous, whereas pre‑trained diffusion backbones (e.g., SD3) encode textual prompts as discrete tokens through their text encoders. As a result, most diffusion‑based editing methods struggle to capture smooth parameter adjustments, limiting their applicability to photographic editing tasks.
>
> 2. Diffusion models lack prior knowledge of camera parameters and fail to capture the corresponding visual effects for each specific parameter.
>
> To address these limitations, we introduce two complementary modules:
>
> - **Continuous Parameter Prompting** interpolates between semantically meaningful anchor embeddings within the continuous text embedding space. This preserves alignment with representation distribution of the pre-trained model while enabling stable, fine-grained, and perceptually coherent control over camera parameters, as demonstrated (see Sec. 5.2).
> - **Parameter-Aware Modulation** moulates the feature inside the diffusion transformer to explicitly capture spatial and global effects that text embeddings alone cannot represent. It applies spatial warping to model geometry/focus changes and channel‑wise scaling to account for exposure and intensity variations.
>
> By integrating these two modules, our framework enables stable, realistic, and continuous camera parameter editing. We will refine Sec. 1 to clarify these connections.
>
> >W3: Request for more discussion on the dataset, including the pros and cons of data sources, the effect of data ratios, and the influence of scale.
>
> Real paired data provide faithful camera effects and realistic details, helping the model learn real parameter-dependent visual cues. However, collecting large-scale paired real data is challenging, particularly for diverse conditions such as animals or uncontrolled outdoor scenes. Synthetic data, on the other hand, enables broader coverage of parameter values and scene diversity while ensuring precise annotations.
>
> We evaluate the impact of these data sources through an ablation study (Table R3-1).
> - Training only on real data (~7k) results in lower perceptual quality and parameter accuracy due to the limited diversity of real scenes and insufficient parameter coverage.
> - Adding synthetic data, with a 1:1 synthetic-to-real ratio (~15k), consistently outperforms training with real data alone. However, the effect is limited by the overall dataset size.
> - Training only on synthetic data (~40k) improves the image content and parameter control due to its large quantity. However, its visual quality is limited due to the lack of guidance from real captured images.
> - Full CamEdit50K, by combining both available synthetic and real data, achieves better parameter control and visual quality. This improvement is driven by both the quantity of synthetic data and the quality of the real data. As shown in Figure 6 of the main paper, when compared with the rendering-based method (BokehMe) used for our synthetic data, our CamEdit significantly improves visual results, reflecting the effectiveness of combining real and synthetic data.
>
> These findings demonstrate that both the diversity and scale of the dataset are critical for maximizing performance. We will incorporate these discussions and the ablation results into the revised manuscript.
>
> Table R3-1: Performance of CamEdit with different synthetic-to-real data ratios.
>
> | Task|           **Aperture** |          |          |           **Focal Plane** |          |          |           **Shutter Speed** |          |             |
> |-------------------------|-----------------------------------:|---------:|---------:|-------------------------------------:|---------:|---------:|---------------------------------------:|---------:|---------:|
> |   Synthetic:Real | NIQE ↓                            | DINO ↑   | Error ↓  | NIQE ↓                              | DINO ↑   | Error ↓  | NIQE ↓                                | DINO ↑   | Error ↓  |
> | 0 : 1     | 4.05         | 0.76  | 1.76    | 4.81            | 0.75  | 0.23    | 4.21               | 0.85  | 0.14    |
> | 1 : 1            | 3.89         | 0.79  | 0.91    | 4.76            | 0.80  | 0.21    | 4.17               | 0.89  | 0.11    |
> | 1 : 0           | 3.91         | 0.82  | 0.66   | 4.93            | 0.76  | 0.15    | 4.19               | 0.87  | 0.12    |
> | **CameEdit50K** | **3.17** | **0.84** | **0.57** | **4.51** | **0.83** | **0.13** | **4.12** | **0.91** | **0.09** |
>
> > W4: Concern about evaluation reliability due to the use of GPT-generated samples.
>
> We clarify that our evaluation is not based on GPT‑generated samples. All quantitative experiments are conducted on real‑captured images. In addition, we include both a human user study and benchmarks on real datasets with ground‑truth captures to further validate the reliability of our results.
>
> 1. **Benchmarks on Real Datasets (Table R3‑2).**
> We benchmark CamEdit on three publicly available real datasets containing ground‑truth captures from digital cameras and photographic lenses: VABD [1] for aperture, EBB Val [2] for focal plane, and SICE [3] for shutter speed. As shown in Table R3‑2, CamEdit outperforms both image editing methods fine‑tuned on CamEdit50K (UltraEdit*) and rendering‑based approaches (DrBokeh, CLODE) across SSIM and LPIPS for all tasks.
>
> 2. **User Study (Table R3‑3).**
> We conduct a user study with 15 participants who have expertise in photography. Each participant evaluates 20 image sets per task across all three tasks, scoring photographic realism, parameter accuracy, and content preservation on a 0–4 scale. As shown in Table R3‑3, CamEdit consistently achieves the highest scores across all metrics, confirming that our results remain valid under human evaluation.
>
> These results demonstrate that our evaluation is grounded in both real-world data and human judgments, ensuring the reliability of our reported performance.
>
> Table R3-2: Quantitative comparison of CamEdit on three real datasets (VABD for Aperture, EBB for Focal Plane, and SICE for Shutter Speed).
> | Task (Dataset)           | Aperture (VABD)             | Focal Plane (EBB Val)           | Task (Dataset)                   | Shutter Speed (SICE)            |
> |---------------------------|------------------------------|----------------------------------|-----------------------------------|----------------------------------|
> | Method                    | SSIM ↑ / LPIPS ↓            | SSIM ↑ / LPIPS ↓                | Method                            | SSIM ↑ / LPIPS ↓                |
> | DrBokeh                   | 0.79 / 0.27                 | 0.79 / 0.24                     | CLODE                             | 0.84 / **0.17** |
> | UltraEdit*                | 0.71 / 0.34                 | 0.73 / 0.31                     | UltraEdit*                        | 0.74 / 0.25                     |
> | **CamEdit (Ours)** | **0.82** / **0.26** | **0.80** / **0.22** | **CamEdit (Ours)** | **0.85** / 0.18                     |
>
> Table R3-3: User Study Scores (0–4 scale) on Photographic Realism, Parameter Accuracy, and Content Preservation
>
> | Task           | Metric                  | UltraEdit / SuperEdit / UltraEdit* / CamEdit  |
> |----------------|-------------------------|-----------------------------------------------|
> | Aperture       | Realism ↑                 | 2.26 / 2.70 / 3.42 / **3.88** |
> |                | Accuracy ↑               | 1.36 / 1.02 / 3.18 / **3.76** |
> |                | Preservation ↑           | 2.10 / 2.76 / 3.38 / **3.87** |
> | Focal Plane    | Realism ↑                | 2.17 / 3.11 / 3.31 / **3.75** |
> |                | Accuracy ↑               | 1.30 / 1.58 / 2.46 / **3.71** |
> |                | Preservation ↑           | 1.80 / 2.83 / 3.14 / **3.81** |
> | Shutter Speed  | Realism ↑                | 2.22 / 3.39 / 3.58 / **3.84** |
> |                | Accuracy ↑               | 1.76 / 1.58 / 2.28 / **3.80** |
> |                | Preservation ↑           | 2.00 / 3.26 / 3.52 / **3.94** |
>
>
> [1] Andrey Ignatov, et al. AIM 2020 Challenge on Rendering Realistic Bokeh. ECCVW, 2020.
>
> [2] Kang Chen, et al. Variable Aperture Bokeh Rendering via Customized Focal Plane Guidance. arXiv, 2024.
>
> [3] Jianrui Cai, et al. Learning a Deep Single Image Contrast Enhancer from Multi-Exposure Images. TIP, 2018.
>
> > L: Request to discuss the limitations of the work.
>
> As mentioned in Section 6 of the main paper, our work already covers some camera parameters, but we plan to extend the approach to include more parameters and editing tasks in future work. We will explicitly outline these plans and limitations in the revised manuscript.

---

> ### Author Response · Authors · 2025-08-04
> **To Reviewer DyLt**
>
> Dear Reviewer DyLt,
>
> We sincerely appreciate your positive feedback, and have addressed your concerns accordingly. As the second review phase nears its end, please let us know if anything remains unclear or if further clarification would be helpful.

---

> > ### Comment · Reviewer_DyLt · 2025-08-05
> > **Response to the rebuttal**
> >
> > Thanks for your rebuttal, most of my concerns have been addressed.

---

> > > ### Author Response · Authors · 2025-08-05
> > > **To Reviewer DyLt**
> > >
> > > We appreciate your review of our response and are grateful for your acknowledgment of our efforts to address the concerns raised.

---

### Official Review · Reviewer_ju7m · 2025-06-30

**Clarity:** 2
**Significance:** 3
**Originality:** 2
**Rating:** 4
**Confidence:** 4

**Summary:**

The paper proposes a diffusion-based framework, CamEdit, for photorealistic image editing that enables continuous control of camera parameters, i.e., aperture, focal plane, and shutter speed. The model is adapted from a pre-trained Stable Diffusion 3 and contains two strategies: (1) a continuous parameter prompting mechanism that interpolates embeddings for smooth parameter transitions, and (2) a parameter-aware modulation (PAM) module that adapts spatial and channel-wise features to reflect physical camera effects. To support training and evaluation, they construct CamEdit50K, a new dataset combining real and synthetic image pairs with annotated camera settings. Experiments demonstrate that CamEdit achieves state-of-the-art performance in visually consistent, parameter-controlled edits, surpassing both diffusion-based baselines and classical rendering methods.

**Questions:**

Please see the weaknesses section.

**Ethical Concerns:**

["NO or VERY MINOR ethics concerns only"]

**Final Justification:**

The rebuttal presents additional results to analyze the dataset curation and the design of the PAM module, providing better support for the claims made in the paper. I increased the score from 3 to 4.

**Limitations:**

The checklist claims that the limitations and potential negative social impact are discussed in Section 6, but that paragraph is too vague and does not provide thorough descriptions.

**Quality:**

2

**Strengths And Weaknesses:**

**Strengths**

- The paper addresses an interesting and practical problem of photorealistic image editing with continuous camera parameter control, which is relatively underexplored.

- A new dataset, CamEdit50K, is introduced, providing paired real and synthetic images annotated with continuous camera settings, which is valuable for training and benchmarking camera-aware editing models.

- The proposed method presents a reasonable and efficient approach to injecting controllability over continuous camera parameters while largely preserving the pre-trained diffusion model’s weights.


**Weaknesses**

- Although the dataset is claimed as one of the main contributions, the paper lacks a thorough analysis of the dataset’s factors, such as how incorporating synthetic data impacts learning, or how the accuracy of estimated camera parameters from real images affects the editing performance of the trained model. Furthermore, it remains unclear whether the design choice of varying only one parameter at a time in synthetic data is deliberate, and whether including examples with simultaneous variations in multiple parameters might help disentangle control and improve learning.

- The clarity of Section 4, particularly Section 4.2, is insufficient. The two components of the parameter-aware modulation module are not visually depicted in the main method figure, and the terminology is somewhat confusing. For instance, it is unclear why average pooling operations are labeled as “Down” instead of using a standard term like “AvgPool.” While grid_sample is a common function in PyTorch, the authors should explicitly describe its operation and role to ensure clarity for readers unfamiliar with its mechanics.

- The paper does not provide a reasonable baseline comparison for the design of the Parameter-Aware Modulation module. A straightforward alternative, such as directly injecting continuous camera parameter features into the timestep embeddings with zero initialization, should be compared to demonstrate the advantages of the proposed approach.

----

**Conclusion**

The paper tackles an interesting and practical problem and achieves reasonable performance; however, it lacks analysis of the dataset, which is presented as a core contribution, lacks reasonable baselines for the continuous camera parameter injection, and requires significant improvements in the clarity of Section 4.

---

> ### Author Rebuttal · Authors · 2025-07-31
>
> We sincerely thank you for your review. Below are your concerns and our corresponding responses:
>
> > W1: Analysis of dataset composition and the accuracy of estimated real-image parameters.
>
> We conduct two sets of ablation studies to address these concerns.
>
> **1. Impact of CamEdit50K composition (Table R2-1).**
> We train models on only real data, only synthetic data, and the full CamEdit50K dataset.
> - Training only on real data (~7k) results in lower perceptual quality and parameter accuracy due to the limited diversity of real scenes and insufficient parameter coverage.
>
> - Training only on synthetic data (~40k) achieves more consistent parameter estimation due to its large quantity. However, its visual quality is limited due to the lack of guidance from real captured images.
>
> - Full CamEdit50K, by combining both available synthetic and real data, achieves better parameter control and visual quality. This improvement is driven by both the quantity of synthetic data and the quality of the real data. As shown in Figure 6 of the main paper, when compared with the rendering-based method (BokehMe) used for our synthetic data, our CamEdit significantly improves visual results, reflecting the effectiveness of combining real and synthetic data.
>
> Table R2-1: Performance of CamEdit with different dataset composition.
>
> | |           Aperture            |          |          |           Focal Plane           |          |          |           Shutter Speed           |          |             |
> |-------------------------|-----------------------------------:|---------:|---------:|-------------------------------------:|---------:|---------:|---------------------------------------:|---------:|---------:|
> |  Datset  | NIQE ↓                            | DINO ↑   | Error ↓  | NIQE ↓                              | DINO ↑   | Error ↓  | NIQE ↓                                | DINO ↑   | Error ↓  |
> | Only on synthetic data       | 3.91|	0.82|	0.66	    | 4.93            | 0.76  | 0.15    | 4.19               | 0.87  | 0.12    |
> | Only on real data     | 4.05         | 0.76  | 1.76    | 4.81            | 0.75  | 0.23    | 4.21               | 0.85  | 0.14    |
> | **CamEdit50K**      | **3.17**         | **0.84**  | **0.57**    | **4.51**            | **0.83**  | **0.13**    | **4.12**               | **0.91**  | **0.09**    |
>
> **2. Effect of estimated camera parameters (Table R2-2).**
> We compare models trained with synthetic data combined with either only EXIF-annotated real data, only estimated-parameter real data, or the full dataset.
>
> - Training on estimated-parameter data improves perceptual quality relative to EXIF-only data, as the estimated subset spans more diverse scenes. Real data with estimated parameters also contribute to improvements in both the visual quality and parameter control.
> - The full dataset achieves the best overall performance, confirming the accuracy of our estimation method. Estimated parameters allow the inclusion of more real paired data into the dataset, further enhancing the model’s performance.
>
> These results demonstrate that both real and synthetic data are essential for optimal performance, and our estimated camera parameters improve coverage and help the model generalize, ensuring reliable editing quality.
>
> Table R2-2: Impact of estimated vs. EXIF camera parameters on editing performance in aperture task.
> | Datset     | NIQE ↓ | DINO ↑ | Error ↓ |
> |--------------------|--------|--------|--------------------|
> |  Synthetic data + Only EXIF          | 3.43   | 0.82   | 0.60               |
> |  Synthetic data + Only Estimated     | 3.31   | 0.83   | 0.61               |
> | **CamEdit50K**      | **3.17**   | **0.84**   | **0.57**               |
>
> >  W1: Design choice of varying one parameter at a time vs. multiple parameters.
>
> We conduct an experiment where the model is trained to adjust all three parameters simultaneously. In this experiment, we double the number of training parameters and include 5,000 synthetic images with multi-parameter variations. As shown in Table R2‑3, the unified model demonstrates accuracy and output quality comparable to single-parameter control, confirming its ability to effectively manage multi-parameter adjustments. We will incorporate these updated results and provide further clarification on this capability in the revised manuscript.
>
> Table R2-3: Performance of CamEdit with single-parameter vs. joint multi-parameter control, evaluated across 200 images with various parameter settings.
>
> | Task             |           **Aperture**            |          |          |           **Focal Plane**           |          |          |           **Shutter Speed**           |          |          |
> |--------------------|-----------------------------------:|---------:|---------:|-------------------------------------:|---------:|---------:|---------------------------------------:|---------:|---------:|
> |                    | NIQE ↓                            | DINO ↑   | Error ↓  | NIQE ↓                              | DINO ↑   | Error ↓  | NIQE ↓                                | DINO ↑   | Error ↓  |
> | **CamEdit (Single-Parameter)** | 3.34         | 0.83   | 0.60   | 4.46             | 0.84   | 0.15   | 4.28               | 0.93   | 0.11   |
> | **CamEdit (Joint-Parameter)** | 3.42         | 0.84   | 0.65   | 4.70             | 0.83   | 0.17   | 4.12               | 0.91   | 0.15   |
>
> > W2: Clarity of Section 4.2 and the description of the Parameter-Aware Modulation module.
>
> In the revised version, we will:
>
> 1. Improve the method figure by explicitly depicting the two components of the parameter-aware modulation module and their roles within the diffusion backbone.
>
> 2. Clarify terminology by replacing the shorthand “Down” with the standard term “AvgPool (2×2)” to avoid confusion.
>
> 3. Explain grid_sample with a concise description of its operation and purpose. In our framework, it is used to warp features according to the camera parameter offsets, enabling spatially aligned modulation of the diffusion features.
>
> We will enhance the readability of Section 4, ensuring that the parameter-aware modulation module is easy to understand.
>
> > W3: Baseline comparison for the Parameter-Aware Modulation (PAM) module.
>
> We conduct more ablation studies to demonstrate the effectiveness of our PAM.
>
> - Replace by TimeStep: injecting continuous camera parameter features into the diffusion timestep embeddings with zero initialization.
> - Replace by Conv: injecting parameters by replacing PAM with convolution-based layers that have the same number of parameters as PAM, but without spatial warping or channel-wise scaling.
>
> As shown in Table R2‑4, both baselines underperform compared to PAM. TimeStep injection yields weaker parameter control due to the absence of localized spatial modeling. Conv injection provides partial improvements in parameter consistency but cannot explicitly capture geometric or global appearance variations.
>
> By contrast, PAM, which combines spatial warping and channel‑wise scaling, achieves the best trade‑off between perceptual quality and parameter accuracy. These results confirm that our PAM design is critical for robust and precise parameter‑aware editing.
>
>  Table R2-4: Comparison of PAM vs. simpler baselines in aperture task.
>
> | Method                | NIQE ↓ | DINO ↑ | Error ↓ |
> |-----------------------|--------|--------|--------------------|
> | Replace by TimeStep   | 3.47   | 0.81   | 0.66               |
> | Replace by Conv       | 3.41   | 0.83   | 0.63               |
> | **Full (PAM)**        | **3.17**   | **0.84**   | **0.57**               |
>
> > L: Discussion of limitations and potential social impact in Section 6.
>
> We will expand Section 6 in the revised version to provide a clearer discussion. Our work already covers a diverse set of camera parameters, and in future work we plan to extend the approach to more parameters and editing tasks. Moreover, we will also discuss the societal risks, especially wrongful image manipulation, and suggest controls like watermarking and post‑processing detection methods.

---

> > ### Comment · Reviewer_ju7m · 2025-08-04
> >
> > Thank you for the thorough additional results. Given the new analyses and ablation studies on dataset curation and the PAM module, I'm more convinced by the paper's contributions and will increase my score from 3 to 4.

---

> > > ### Author Response · Authors · 2025-08-04
> > > **To Reviewer ju7m**
> > >
> > > We sincerely thank you for carefully reviewing our response. We really appreciate your recognition of our effort to address the concerns raised, and we are grateful for your support for the paper's acceptance.

---

> ### Author Response · Authors · 2025-08-04
> **To Reviewer ju7m**
>
> Reviewer ju7m,
>
> We sincerely appreciate your feedback and have addressed your concerns accordingly. As the second review phase is nearing its end, we kindly ask if there are any remaining questions or clarifications we can assist with.

---

### Official Review · Reviewer_rxHt · 2025-07-01

**Clarity:** 3
**Significance:** 3
**Originality:** 2
**Rating:** 4
**Confidence:** 4

**Summary:**

CamEdit: Continuous Camera Parameter Control for Photorealistic Image Editing presents a diffusion generative model to modify images based on camera parameters using text prompts. The task is controlled photorealistic conditioned generation to simulate the change of aperture, focal plane and shutter speed parameters using continuous physical quantities. To achieve this a continuous parameter embedding learns to interpolate over discrete camera tokens that are merged with the original prompt tokens. Additionally, features are modulated after the self-attention layers of the DIT conditioned on the camera parameters from the input prompt.
For training of the newly added modules and fine-tuning of the overall model using LoRA, a new dataset is generated using images from various photographic datasets and synthetic data. The photographic data is annotated using a custom annotation pipeline for camera parameters. In the quantiative evaluation the method reaches state-of-the-art in several reference-free metrics and an evaluation using GPT-4o.

**Questions:**

- How does the model output compare to ground truth footage shot with a digital camera and photographic lens, for example? Do the parameter scales make sense in this case and how are additional lens artifacts like aberrations, distortion etc. handled by this approach? Does the diffusion model know about those? Or are they getting lost due to the limitations in the data generation process? Some comparison would be critical to justify the claim of photorealism here, I think. This could also be achieved with simulated data.

- What camera model is the basis of the problem statement? It might be better to introduce a common camera model that explains all the mentioned effects early on in the method section (move some parts from the supplements?)

- What are the advantages of textual input? Why not using a different e.g. slider based input which might be more intuitive for users of photo editing software?

- An important part of the work is the data generation that builds a nice pipeline for image annotation of camera related parameters. It would be good to include the image sources in the main paper here (instead of the supplementary) I think as this lead to some questions that weren't answered by a reference, for example.

- How consistent are the generations? An evaluation over multiple generations might be interesting here. At least reporting the standard deviations in the tables could give some hint.

**Ethical Concerns:**

["NO or VERY MINOR ethics concerns only"]

**Final Justification:**

Thank you for the detailed answers. Overall, many questions have been addressed adequantely and I'm happy to increse the final rating in line with the other reviewers.

With the added evaluations, including the user study, and the refinements of the text as proposed in the rebuttal I can see the paper as part of the conference program. The method outperforms prior work and the authors manage to show that also a multi-parameter model variant can be trained which would enable broader application.

**Limitations:**

Model limitations are adequately discussed, the societal impact of manipulations of photographic material should probably be added.

**Paper Formatting Concerns:**

No concerns.

**Quality:**

2

**Strengths And Weaknesses:**

Strength:
- The paper convincingly introduces a method to improve the controllability in generative image manipulation tasks that are related to changing camera parameters. This is a relevant task for photography and design applications and might also help simulation tasks for autonomuous and robotic systems.
- The dataset generation pipeline shows some effort and the dataset could be useful for the research community.
- The method achieves good scores in all applied metrics.


Weaknesses:
- In the context of the related works and the methods problem setting, works like ConceptSliders [1] would be relevant, too.
- While relative adjustments look plausible in the qualitative analysis and work well in no-reference metrics, some analysis against actually shot ground truth is missing, which would be available from any professional digital photo camera.
- In the image editing ecosystem text-based interaction might not be the most efficient way to change image parameters.
- The limitation of only editing one parameter at a time is also limiting the applicability.
- The shutter parameter seems to work like a gain / exposure adjustment here. While this is one of the effects of changing the shutter time there is also a change in how scene motion is rendered. This is missing from the analysis and the shown results. E.g. if I increase the shutter speed I would expect to see more motion blur in the scene (from moving objects and potentially camera shake). This weakness also seems to be reflected in the lower photorealism score in the GPT-5o evaluation.  Additional image changes like noise patterns, aberrations and distortions are not discussed in the paper, although they have quite some impact on the perceived photorealism of an edit.
- While ChatGPT-4o might be a good proxy, a user study with real human subjects would be interesting and is missing in the evaluation.
- Camera parameter modification does not work in all directions equally well (as some destructive effects cannot be undone by the model). This might be out of scope for a single work but would be an interesting trajecory for further research.

Small typos:
- Figure 5 caption: "Additional" -> "Additionally"
- BrokehMe instead of BokehMe, p.6, l.216

1: Gandikota et al. Concept Sliders: LoRA Adaptors for Precise Control in Diffusion Models. ECCV 2024.

---

> ### Author Rebuttal · Authors · 2025-07-31
>
> We sincerely thank you for your review. Below are your concerns and our corresponding responses:
> >W1: Discussion of related work such as ConceptSliders.
>
> ConceptSliders is designed for common editing tasks such as changing human age and expressions or performing style transfer. However, it is not directly applicable to photorealistic image editing. However, our work proposes a photorealistic editing framework that supports tasks such as focus swapping, aperture adjustment, and exposure modification. We will include this discussion in the related work section.
>
> > W2 & Q1: Evaluation against ground-truth captures from professional digital cameras.
>
> We evaluate CamEdit on three publicly available datasets with ground-truth captures from digital cameras: VABD [1] for aperture, EBB Val [2] for focal plane, and SICE [3] for shutter speed.  As shown in Table R1‑1, CamEdit outperforms both fine-tuned image editing methods (UltraEdit*, retrained on CamEdit50K) and rendering-based approaches (DrBokeh, CLODE) across all tasks, achieving superior SSIM and LPIPS scores.
>
> Table R1-1: Quantitative comparison of CamEdit on three real datasets (VABD for Aperture, EBB for Focal Plane, and SICE for Shutter Speed).
> | Task (Dataset)           | Aperture (VABD)             | Focal Plane (EBB Val)           | Task (Dataset)                   | Shutter Speed (SICE)            |
> |---------------------------|------------------------------|----------------------------------|-----------------------------------|----------------------------------|
> | Method                    | SSIM ↑ / LPIPS ↓            | SSIM ↑ / LPIPS ↓                | Method                            | SSIM ↑ / LPIPS ↓                |
> | DrBokeh                   | 0.79 / 0.27                 | 0.79 / 0.24                     | CLODE                             | 0.84 / **0.17** |
> | UltraEdit*                | 0.71 / 0.34                 | 0.73 / 0.31                     | UltraEdit*                        | 0.74 / 0.25                     |
> | **CamEdit (Ours)** | **0.82** / **0.26** | **0.80** / **0.22** | **CamEdit (Ours)** | **0.86** / 0.18                     |
>
> [1] Andrey Ignatov et al. AIM 2020 Challenge on Rendering Realistic Bokeh. ECCVW, 2020.
> [2] Kang Chen et al. Variable Aperture Bokeh Rendering via Customized Focal Plane Guidance. arXiv, 2024.
> [3] Jianrui Cai et al. Learning a Deep Single Image Contrast Enhancer from Multi-Exposure Images. TIP, 2018.
>
> > W3&Q3: Advantages of text-based input vs. other controls.
>
> - Embedding camera parameters as textual inputs leverages the T2I diffusion model’s prior knowledge, improving the model’s understanding of the parameters and performing more accurate camera parameter editing, as demonstrated in Section 5.3 and Figure 7 of the main paper, compared to direct parameter mapping or linear approaches.
>
> - Moreover, the digital parameters of textual inputs can be directly mapped to slider‑based or other controls, with the editing software’s UI adapting to the user’s choice.
>
> > W4: Concern about editing only one parameter at a time.
>
> We conduct an experiment where the model is trained to adjust all three parameters simultaneously. In this experiment, we double the number of training parameters and include 5,000 synthetic images with multi-parameter variations. As shown in Table R1‑2, the unified model demonstrates accuracy and output quality comparable to single-parameter control, confirming its ability to effectively manage multi-parameter adjustments. We will incorporate these updated results and provide further clarification on this capability in the revised manuscript.
>
> Table R1-2: Performance of CamEdit with single-parameter vs. joint multi-parameter control, evaluated across 200 images with various parameter settings.
>
> | Task             |           Aperture            |          |          |           Focal Plane           |          |          |           Shutter Speed           |          |          |
> |--------------------|-----------------------------------:|---------:|---------:|-------------------------------------:|---------:|---------:|---------------------------------------:|---------:|---------:|
> |                    | NIQE ↓                            | DINO ↑   | Error ↓  | NIQE ↓                              | DINO ↑   | Error ↓  | NIQE ↓                                | DINO ↑   | Error ↓  |
> | **CamEdit (Single-Parameter)** | 3.34         | 0.83   | 0.60   | 4.46             | 0.84   | 0.15   | 4.28               | 0.93   | 0.11   |
> | **CamEdit (Joint-Parameter)** | 3.42         | 0.84   | 0.65   | 4.70             | 0.83   | 0.17   | 4.12               | 0.91   | 0.15   |
>
> > W6: The user study with real human subjects.
>
> We conducted a user study with 15 participants who have expertise in photography. Each participant evaluated 20 image sets per task across all three tasks, rating photographic realism, parameter accuracy, and content preservation on a 0–4 scale.
> As shown in Table R1‑3, CamEdit consistently achieves the highest scores across all metrics, confirming the validity of our results under human evaluation.
>
>
> Table R1-3: User Study Scores (0–4 scale) on Photographic Realism, Parameter Accuracy, and Content Preservation
>
> | Task           | Metric                  | UltraEdit / SuperEdit / UltraEdit* / CamEdit  |
> |----------------|-------------------------|-----------------------------------------------|
> | Aperture       | Realism ↑                 | 2.26 / 2.70 / 3.42 / **3.88** |
> |                | Accuracy ↑               | 1.36 / 1.02 / 3.18 / **3.76** |
> |                | Preservation ↑           | 2.10 / 2.76 / 3.38 / **3.87** |
> | Focal Plane    | Realism ↑                | 2.71 / 3.11 / 3.31 / **3.75** |
> |                | Accuracy ↑               | 1.30 / 1.58 / 2.46 / **3.71** |
> |                | Preservation ↑           | 1.80 / 2.83 / 3.14 / **3.81** |
> | Shutter Speed  | Realism ↑                | 2.22 / 3.39 / 3.58 / **3.84** |
> |                | Accuracy ↑               | 2.76 / 1.58 / 2.28 / **3.80** |
> |                | Preservation ↑           | 2.00 / 3.26 / 3.52 / **3.94** |
>
> > W5&Q1: Concern about the lack of motion blur modeling for shutter speed and omission of lens artifacts (e.g., distortions).
>
> 1. **On motion blur and shutter speed:**
> - CamEdit models shutter speed via exposure adjustment, following standard single-image shutter modeling practices (e.g., PhotoGen [4]), which are widely used in computational photography.
> - Motion-induced effects like blur from object or camera movement require multi-frame data and are beyond the scope of this single-image editing framework.
> - The slightly lower GPT‑4o score is mainly due to brightness changes, as the model tends to penalize low-light images. Nonetheless, as shown in Table R3‑2 and R3‑3, CamEdit consistently outperforms existing methods in both human studies and comparisons with real ground-truth data.
>
> 2. **On lens artifacts, parameter scales, and real-camera comparisons:** Complex lens artifacts such as chromatic aberrations and geometric distortions are not controlled by camera parameters, and are therefore outside the scope of this work.
>
> [4] Yu Yuan et al. Generative photography: Scene-consistent camera control for realistic text-to-image synthesis. CVPR, 2025.
>
> > W7: Unequal effectiveness of camera parameter adjustments, as some destructive effects cannot be undone.
>
> As noted in Lines 34–36 and 89–94, CamEdit is consistent with existing rendering-based methods and physical camera models, and is designed for precise camera parameter adjustments. When input image details are lost, reversibility is not feasible for any current editing framework, as the missing information cannot be recovered. Addressing such cases would require dedicated image restoration strategies, which we identify as a promising direction for future research.
>
> > Q2&Q4: Clarifying the underlying camera model and data pipeline.
>
> In the revised version, we will integrate the camera model into the main method section for a unified explanation of all referenced effects, and move the data generation process and image sources from the supplementary into the main text for clarity.
>
> > Q5: Consistency of generations
>
> We evaluate the consistency of CamEdit across different parameter settings using 200 images per task. For each image, we compute the mean and standard deviation of the DINO similarity between the generated outputs and their corresponding inputs, across five different parameter settings for the same image.
>
> CamEdit demonstrates high consistency with low variation:
>
> **Aperture:** 0.83 ± 0.09
>
> **Focal Plane:** 0.84 ± 0.11
>
> **Shutter Speed:** 0.93 ± 0.06
>
> For aperture and focal plane, we further analyzed foreground and in-focus regions separately, where the DINO standard deviations drop to 0.04 and 0.05, respectively. These results indicate that CamEdit’s edits are stable, with negligible changes outside the parameter-driven effects.
>
> Qualitative results (Figure 1, Figure 5 in the main paper; Figure S3, Figure S5 in the supplementary) also show that CamEdit modifies only the target camera attributes without affecting unrelated scene content, further confirming the consistency of CamEdit.
>
> > L: The societal impact of photographic manipulations.
>
> In the revised version, we will explicitly discuss potential societal impacts, including the risk of misuse for deceptive image manipulation, and outline mitigation strategies such as watermarking and edit-detection mechanisms.
>
> > About Typos.
>
> We will correct all typos in the revision.

---

> > ### Comment · Reviewer_rxHt · 2025-08-05
> >
> > Thank you for the detailed answers. Overall, many questions have been addressed adequantely and I'm happy to increse the final rating in line with the other reviewers.
> >
> > W1: The reference is mostly connected to the aspect of continuous parameters as user input for conditioning and I still see it as related.
> > W5 / Q1: The effect of aberrations is dependent on the aperture opening, for example. But I agree that this might be out of scope of the work.
> >
> > With the added evaluations and the proposed refinements of the text I can see the paper as part of the conference program.

---

> > > ### Author Response · Authors · 2025-08-06
> > > **To Reviewer rxHt**
> > >
> > > Thank you for reviewing our response and for supporting our paper for the conference program.
> > >
> > > W1: Regarding the reference to user input for continuous editing, we agree that fine-grained methods like ConceptSliders are closely related to the approach in our paper. We will include this, along with references to related methods, in the revised version.
> > >
> > > W5/Q1: As for the effect of aberrations, we plan to explore more complex camera models in future research.
> > >
> > > I really appreciate your willingness to increase the final rating in line with the other reviewers. Based on your suggestions, we will incorporate the proposed refinements and additional evaluations in the revised manuscript.

---

> ### Author Response · Authors · 2025-08-04
> **To Reviewer rxHt**
>
> Dear Reviewer rxHt,
>
> We sincerely appreciate your feedback and have addressed your concerns accordingly. As the second review phase is coming to a close, we kindly ask if there are any remaining questions or clarifications we can assist with.

---

> > ### Comment · Area_Chair_RuYx · 2025-08-05
> >
> > Dear reviewer rxHt,
> >
> > The authors have provided extensive ablations on real-world data, metrics and comparison to baselines. As the Author-Reviewer discussion is coming to an end, we appreciate your participation in the discussion. We kindly remind you that the reviewers must participate in the discussion, that might otherwise result in "InsufficientReview" flag.
> >
> > Did the the authors answer your concerns? Does it change your score?
> >
> > -- Your AC

---

### Note · Authors · 2025-08-12

**Final Remarks to the AC**

We sincerely appreciate the AC and reviewers for their time and thoughtful feedback, as well as their active, constructive engagement during the discussion phase. We are encouraged by the positive assessments from all reviewers, which highlight the following strengths of our work:

- **Novel research direction** [rxHt, ju7m, DyLt, U816]: The proposed CamEdit addresses an underexplored problem—photorealistic image editing with continuous control over camera parameters—with clear value for both the research community and practical applications.

- **Innovative and effective solution** [rxHt, ju7m, DyLt, U816]: CamEdit enables continuous camera-parameter awareness by interpolating in the text-embedding space. It further modulates features inside the diffusion transformer to explicitly capture parameter-induced spatial and global effects. We also contribute a dedicated CamEdit dataset and a reproducible data-generation pipeline. This approach is both innovative and novel.

- **Impressive results** [rxHt, rwnV, ju7m, U816]: CamEdit delivers parameter-controlled edits with fine-grained detail and strong consistency, effectively addressing the challenge in image editing.

- **Clarity and Readability** [rwnV, DyLt, U816]: Our paper is well-motivated, clearly articulated, and easy to read.

We also thank all reviewers for their insightful and constructive suggestions, which help further improve our paper. In addition to the pointwise responses below, we summarize the major revisions in the rebuttal according to the reviewers’ suggestions.

- **Comparative study on real data and user study**: We provide extensive comparisons on real-captured datasets with ground truth and a user study on reference-free images, further validating CamEdit’s faithful camera-parameter editing and photorealistic edit quality.

- **Joint multi-parameter editing**: We include experiments that jointly edit multiple camera parameters, showing CamEdit’s fine-grained control scales to the multi-parameter setting.

- **Method ablations and clarifications**: We provide detailed component comparisons of CamEdit, ablation variants for parameter-aware modulation and an in-depth analysis of the proposed method.

We believe these additions clarify the method, strengthen the empirical evidence, and address the main concerns raised during the discussion. Thank you for your consideration.

Best,

Authors

---

### Decision · Program_Chairs · 2025-09-17

**Decision:**

Accept (poster)

**Comment:**

The paper introduces camera-aware image editing, allowing continuous manipulation of camera parameters such as aperture, shutter speed and focal plane. The work introduces a number of techniques on how to incorporate these camera parameters into a diffusion backbone. This work also introduces a new datasets CamEdit50k with different camera settings.

The authors addressed the concerns from the reviewers: evaluated CamEdit on three real-world datasets, added dataset ablations on CamEdit50K (synthetic vs real vs combined), expanded the evaluation set and provided model ablations.

The reviewers agree that the paper is recommended for acceptance.